# A phosphoswitch at acinus-serine[437] controls autophagic responses to cadmium exposure and neurodegenerative stress

**Nilay Nandi[1†], Zuhair Zaidi[1†], Charles Tracy[1], Helmut Krämer[1,2*]**

[1]Department of Neuroscience, UT Southwestern Medical Center, Dallas, United States; [2]Department of Cell Biology, UT Southwestern Medical Center, Dallas, United States

**Abstract** Neuronal health depends on quality control functions of autophagy, but mechanisms regulating neuronal autophagy are poorly understood. Previously, we showed that in *Drosophila* starvation-independent quality control autophagy is regulated by acinus (acn) and the Cdk5-dependent phosphorylation of its serine[437] (Nandi et al., 2017). Here, we identify the phosphatase that counterbalances this activity and provides for the dynamic nature of acinus-serine[437] (acn-S437) phosphorylation. A genetic screen identified six phosphatases that genetically interacted with an acn gain-of-function model. Among these, loss of function of only one, the PPM-type phosphatase Nil (CG6036), enhanced pS437-acn levels. Cdk5-dependent phosphorylation of acn-S437 in *nil*[1] animals elevates neuronal autophagy and reduces the accumulation of polyQ proteins in a *Drosophila* Huntington's disease model. Consistent with previous findings that $Cd^{2+}$ inhibits PPM-type phosphatases, $Cd^{2+}$ exposure elevated acn-S437 phosphorylation which was necessary for increased neuronal autophagy and protection against $Cd^{2+}$-induced cytotoxicity. Together, our data establish the acn-S437 phosphoswitch as critical integrator of multiple stress signals regulating neuronal autophagy.

**\*For correspondence:**
helmut.kramer@utsouthwestern.edu

[†]These authors contributed equally to this work

**Competing interest:** The authors declare that no competing interests exist.

## Editor's evaluation

The paper is of broad interest to readers focusing on quality control functions of autophagy contributing towards neuronal health. This work provides substantial new insights into the molecular mechanisms underlying the dynamic nature of the quality control function of autophagy. Building on their previous work, in the current advance the authors identify phosphatase as crucial for controlling a phospho-switch which counteracts a kinase complex dependent phosphorylation. The authors demonstrated this phospho-switch as a key integrator of multiple stress signals including. Overall the findings are interesting and important.

## Introduction

A key process for maintaining cellular fitness is autophagy, here short for macroautophagy (*Fleming and Rubinsztein, 2020; Menzies et al., 2015*). Starvation induces nonselective autophagy which contributes to reclaiming molecular building blocks (*Levine and Kroemer, 2019*). In neurons and other long-lived cells, quality control of proteins and organelles is an additional critical function of autophagy (*Dong et al., 2021; Evans and Holzbaur, 2020; Kroemer et al., 2010*). The importance of the quality control function of starvation-independent basal autophagy was demonstrated by mutations in core autophagy components in mice and flies: cell-type specific loss of Atg5 or Atg7

triggers rapid neurodegeneration (*Hara et al., 2006*; *Juhász et al., 2007*; *Komatsu et al., 2006*) or cardiac hypertrophy (*Nakai et al., 2007*). Moreover, elevated basal autophagy can successfully reduce the polyQ load in models of Huntington's disease or spinocerebellar ataxia type 3 and reduce neurodegeneration (*Bilen and Bonini, 2007*; *Jaiswal et al., 2012*; *Nandi et al., 2014*; *Nandi et al., 2017*; *Ravikumar et al., 2004*). Both modes of autophagy use core autophagy proteins to initiate the generation of isolation membranes (also known as phagophores), promote their growth to autophagosomes, and finally promote their fusion with lysosomes or late endosomes to initiate degradation of captured content (*Mizushima, 2017*). Although the rapid induction of autophagy in response to nutrient deprivation is well described (*Galluzzi et al., 2017*), much less is known about the modulation of basal levels of autophagy in response to cellular stress.

We previously identified acinus (acn) as a regulator of starvation-independent quality control autophagy in *Drosophila* (*Haberman et al., 2010*; *Nandi et al., 2014*; *Nandi et al., 2017*). Acn is a conserved protein enriched in the nucleus and, together with Sin3-associated protein of 18 kDa (Sap18) and RNA-binding protein S1 (RNPS1), forms the Apoptosis and splicing–associated protein (ASAP) complex (*Murachelli et al., 2012*; *Schwerk et al., 2003*). The ASAP complex can regulate alternative splicing by interacting with the exon junction complex, spliceosomes, and messenger ribonucleoprotein particles (*Hayashi et al., 2014*; *Malone et al., 2014*; *Tang et al., 1995*). In mammals and *Drosophila*, acn levels are regulated by its Akt1-dependent phosphorylation which inhibits caspase-mediated cleavage (*Hu et al., 2005*; *Nandi et al., 2014*). Furthermore, acn stability is enhanced by Cdk5-mediated phosphorylation of the conserved serine-437. Acn levels are elevated by the phosphomimetic AcnS437D mutation and reduced by the phosphoinert Acn-S437A (*Nandi et al., 2017*). The stress-responsive Cdk5/p35 kinase complex (*Su and Tsai, 2011*) regulates multiple neuronal functions including synapse homeostasis and axonal transport (*Klinman and Holzbaur, 2015*; *Lai and Ip, 2015*; *McLinden et al., 2012*), in addition to its role in autophagy (*Nandi and Krämer, 2018*; *Shukla and Giniger, 2019*). Phosphorylation-induced stabilization of Acn increases basal, starvation-independent autophagy with beneficial consequences including reduced polyQ load in a *Drosophila* Huntington's disease model and prolonged life span (*Nandi et al., 2014*; *Nandi et al., 2017*). The detailed mechanism by which Acn regulates autophagy is not well understood, but is likely to involve the activation of Atg1 kinase activity as autophagy-related and unrelated functions of Atg1 are enhanced by elevated Acn levels (*Nandi et al., 2014*; *Nandi et al., 2017*; *Tyra et al., 2020*). Identification of Acn in a high-content RNAi screen for genes promoting viral autophagy in mammals (*Orvedahl et al., 2011*) suggests a conserved role in regulating starvation-independent autophagy.

Detailed examination of the cell type-specific changes in levels and phosphorylation of Acn in photoreceptor neurons of developing larval eye discs revealed a highly dynamic pattern (*Nandi et al., 2014*; *Nandi et al., 2017*). This motivated us to investigate the role of serine-threonine phosphatases in counteracting Cdk5/p35 kinase-mediated Acn phosphorylation. In a targeted screen, we identified CG6036, a member of the PPM family of protein phosphatases as critical for controling the phosphoswitch on Acn-S437. PPM-type phosphatases are dependent on $Mg^{2+}$ or $Mn^{2+}$ as cofactor for their activity (*Kamada et al., 2020*). They are not inhibited by the broad-spectrum phosphatase inhibitor okadaic acid, in contrast to phosphoprotein phosphatase (PPP)-type phosphatases and do not require the regulatory subunits characteristic for PPP-type phosphatases. Instead, the PPM family contains additional domains and conserved motifs, which can determine its substrate specificity (*Andreeva and Kutuzov, 2001*; *Shi, 2009*; *Tong et al., 1998*). Intriguingly, PPM family members play a pivotal role in different physiological or pathological processes that are responsive to cellular stress signaling, including regulation of AMPK (*Davies et al., 1995*), Tak1 (*Hanada et al., 2001*), or other mitogen-activated protein (MAP) kinases (*Hanada et al., 1998*; *Maeda et al., 1994*; *Shiozaki et al., 1994*; *Takekawa et al., 1998*).

Our findings indicate that the CG6036 phosphatase, through its effect on Acn phosphorylation, regulates neuronal responses to proteostasis or toxicological stress. We renamed this phosphatase Nilkantha (Nil) meaning 'Blue Throat.' In Hindu mythology, Lord Shiva's throat turned blue upon drinking a poison thereby saving all the living. Similarly, our data indicate that deactivation of the CG6036 phosphatase (Nil) by Cadmium poisoning is critical for coping with the cellular stress it causes.

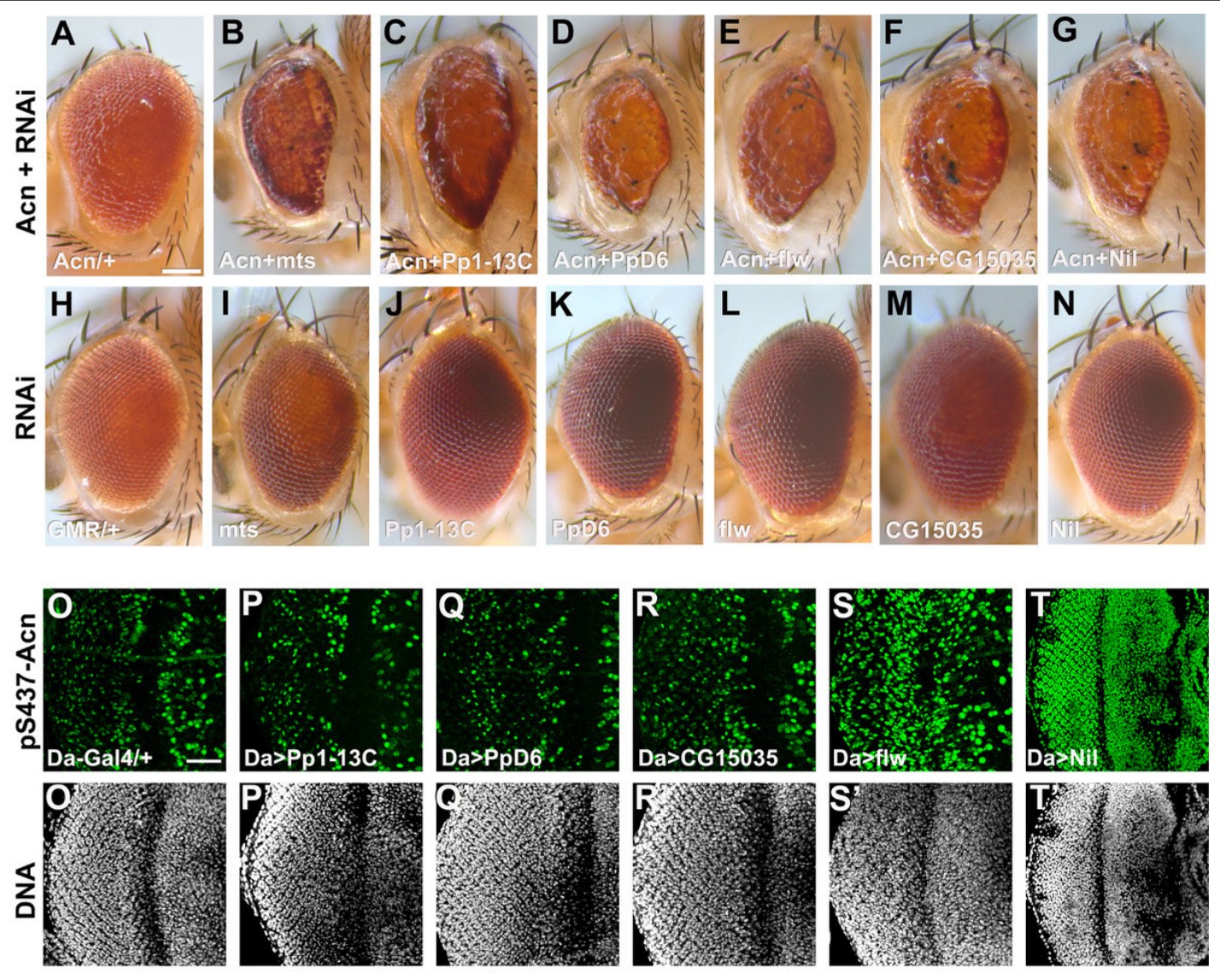

**Figure 1.** A genetic screen identifies Nil as the Acinus-Serine[437] phosphatase. (A–N) Micrographs of eyes in which GMR-Gal4 drives expression of AcnWT (**A**), AcnWT+ microtubule star (mts)–RNAi (**B**), AcnWT+ Pp1-13 C RNAi (**C**), AcnWT+ PpD6 RNAi (**D**), AcnWT+ flapwing (flw) RNAi (**E**), AcnWT+ CG15035 RNAi (**F**), AcnWT+ nil RNAi (**G**), mts–RNAi (**I**), Pp1-13C–RNAi (**J**), PpD6–RNAi (**K**), flw–RNAi (**L**), CG15035–RNAi (**M**), nil–RNAi (**N**), and H represents GMR-Gal4 control. (O–T) Projection of confocal micrographs of larval eye discs stained for pS437-Acn (green) and DNA from Da-Gal4 (**O, O′**), Da-Gal4, Pp1-13C RNAi (**P, P′**), Da-Gal4, PpD6 RNAi (**Q, Q′**), Da-Gal4, CG15035 RNAi (**R, R′**), Da-Gal4, flw RNAi (**S, S′**), Da-Gal4, nil RNAi (**T, T′**). Scale bar in A is 100 µm for A-N, scale bar in O is 40 µm for O-T. Genotypes are listed in *Supplementary file 3*.

The online version of this article includes the following source data and figure supplement(s) for figure 1:

**Source data 1.** Phosphatase RNAi lines were crossed to Da-Gal4 in order to measure knockdown of Phosphatases which are hits in the screen.

**Figure supplement 1.** qPCR analysis of phosphatase genes interacting with Acinus.

## Results

### Nil phosphatase regulates phosphorylation of the conserved serine 437 of Acn

To identify phosphatases responsible for modulating Acn function, we performed a targeted RNAi screen of the 37 non-CTD-type serine-threonine phosphatases encoded in the *Drosophila* genome (*Supplementary file 1*). To test the effect of these phosphatases on Acn function, we used an eye-specific sensitized genetic system. GMR-Gal4-driven expression of wild-type Acinus(UAS-Acn^WT) at

28°C yields a rough-eye phenotype (*Figure 1A*), that is modified by genetic enhancers or suppressors (*Nandi et al., 2014*; *Nandi et al., 2017*). We reasoned that knocking down a phosphatase responsible for dephosphorylating Acn would elevate the levels of phosphorylated Acn and hence stabilize the Acn protein, resulting in an enhancement of the eye roughness induced by UAS-Acn[WT]. Among the serine-threonine phosphatases encoded by the *Drosophila* genome, RNAi lines targeting *CG6036*, *CG15035*, *PpD6*, *Pp1-13C*, *flapwing (flw)*, and *microtubule star (mts)* exhibited enhancement of Acn-induced eye roughness yielding a severely rough and reduced eye (*Figure 1A–G*, *Supplementary file 1*, *Figure 1—figure supplement 1*). By contrast, expression of these RNAi transgenes by themselves did not result in visible eye phenotypes (*Figure 1H–N*, *Supplementary file 1*, *Figure 1—figure supplement 1*).

To test whether these genetic interactions reflect direct effects on the phosphorylation status of Acn, we used a phospho-specific antibody raised against pS437-Acn (*Nandi et al., 2017*). Levels of pS437-Acn were evaluated in eye discs in which these phosphatases had been knocked down using the ubiquitously expressed Da-Gal4 driver. No change in Acn phosphorylation resulted from knockdown of the phosphatases Pp1-13C, PpD6, or CG15035 (*Figure 1O–R*, *Figure 1—figure supplement 1*). By contrast, eye discs with *flw* knockdown displayed an altered pattern of pS437-Acn positive cells (*Figure 1*, *Figure 1—figure supplement 1*). Moreover, knocking down *mts* with Da-Gal4 resulted in larval lethality.

Interestingly, the PP2A phosphatase mts, a member of the Striatin-interacting phosphatase and kinase (STRIPAK) complex, and the PP1 phosphatase flw regulate upstream components of Hippo/Yorkie signaling (*Gil-Ranedo et al., 2019*; *Neal et al., 2020*; *Ribeiro et al., 2010*; *Yang et al., 2012*). Furthermore, Yorkie's growth promoting activity is regulated by Acn activity (*Tyra et al., 2020*). Taken together, this suggests that the strong genetic interactions of Acn with the mts and flw phosphatases might reflect additive effects on Yorkie activity rather than direct effects on Acn phosphorylation.

By contrast, knock down of Nil (CG6036) yielded a dramatic enhancement of Acn phosphorylation at serine[437] compared to wild-type controls (*Figure 1O and T*, *Figure 1—figure supplement 1*). Given this robust increase of Acn phosphorylation, we further explored the role of Nil in regulating Acn function.

Nil is a member of the PPM family of phosphatases characterized by multiple conserved acidic residues (*Figure 2A*) that contribute to a binuclear metal center critical for phosphatase activity (*Das et al., 1996*; *Pan et al., 2013*). To further test the role of the Nil phosphatase in regulating Acn-S437 phosphorylation, we used CRISPR/Cas9 to generate the *nil*[1] deletion allele that eliminates the majority of the conserved phosphatase domain (*Figure 2A*). Antennal discs and larval fat bodies from *nil*[1] wandering larvae displayed a dramatic increase in Acn-S437 phosphorylation compared to wild-type controls (*Figure 2B–E*). A similar robust enhancement of pS437-Acn staining was seen in *nil*[1] mutant eye discs compared to the controls (*Figure 2F and G*). Overexpressing wild-type Nil or the human PPM1B (Protein phosphatase, Mg$^{2+}$/Mn$^{2+}$dependent 1B) homolog of Nil restored phosphatase activity in *nil*[1] mutant eye discs (*Figure 2H and I*).

Multiple sequence alignment with other PPM-type phosphatases pointed to aspartate-231 of Nil as an acidic residue critical for metal binding and phosphatase activity (*Kamada et al., 2020*). Mutation of this aspartate residue to asparagine generated the Nil[D231N] point mutant; its expression in *nil*[1] mutant eye discs failed to restore phosphatase activity (*Figure 2J*). Moreover, the rough eye phenotype induced by Acn overexpression using the GMR-Gal4 driver was suppressed by coexpression of wild-type Nil, but not Nil[D231N] (*Figure 2—figure supplement 1A-F*). Additionally, overexpression of wild-type Nil, but not the inactive Nil[D231N] mutant, in larval eye discs reduced pS437-Acn levels compared to GMR-Gal4 only controls (*Figure 2—figure supplement 1G-I*).

## Nil phosphatase localizes to the nucleus and to endolysosomal Compartments

To gain insights into how this phosphatase can regulate Acn phosphorylation and function, we generated the *nil*[Ty1-G4] allele expressing Ty1-tagged Nil phosphatase and Gal4 under control of endogenous nil promoter (*Figure 2A*, *Figure 2—figure supplement 2O*). Expression of UAS-mCD8 GFP using this Gal4 driver revealed wide-spread low-level expression in the adult brain, eye discs and larval fat bodies, and elevated expression in testis (*Figure 2—figure supplement 2F-N*) consistent with previously reported expression levels (*Brown et al., 2014*).

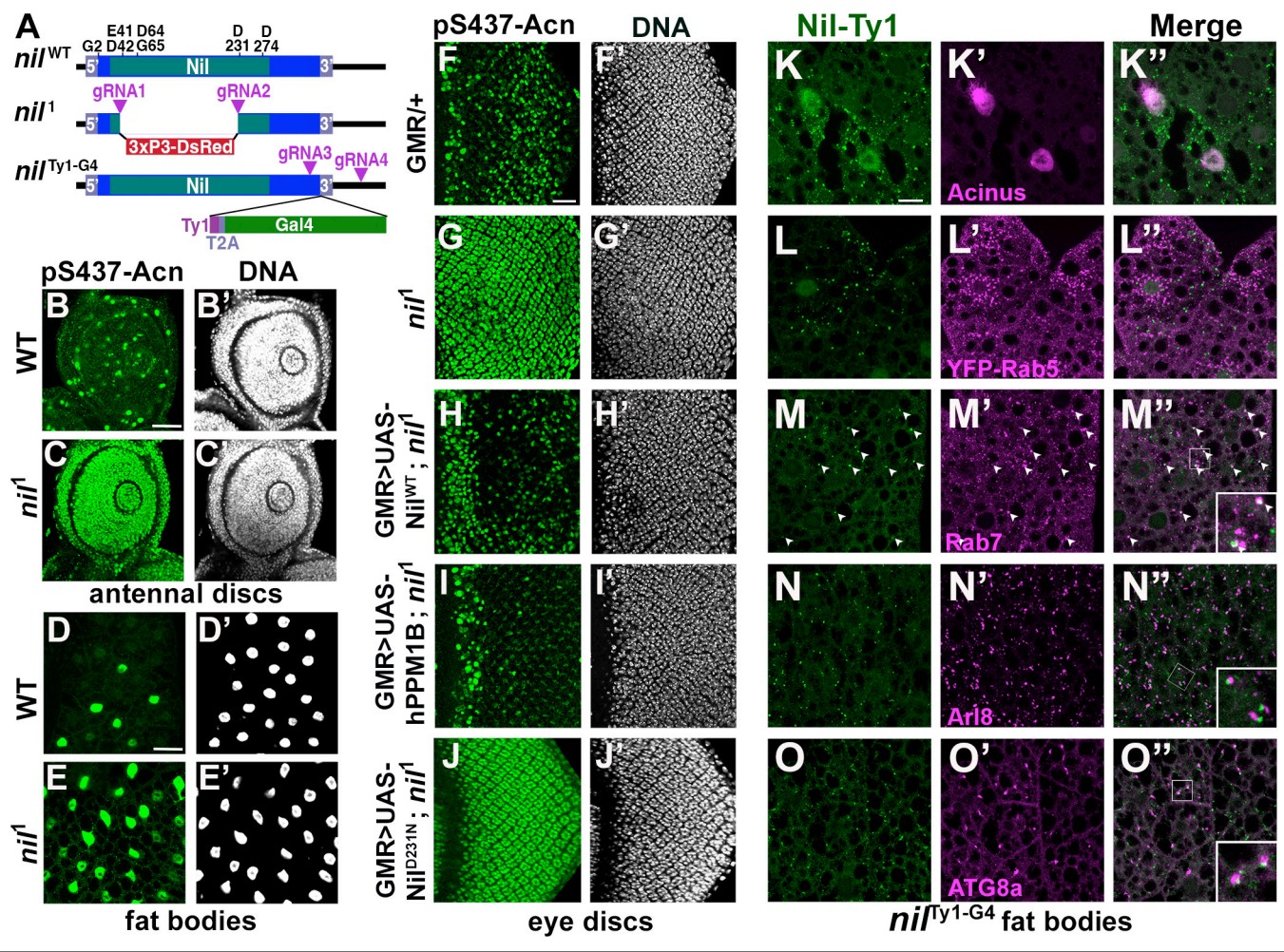

**Figure 2.** Nil loss and gain of function regulates Acinus-Serine$^{437}$ phosphorylation. (**A**) Diagram depicting the Nil$^{WT}$ protein with amino acids highly conserved in the PPM family of phosphatases, the $nil^1$ allele with its 3xP3-DsRed insertion, and the multicistronic $nil^{Ty1-G4}$ allele with Ty1 tag and T2A-coexpressed Gal4. (**B–E**) Projection of confocal micrographs of larval antennal discs (**B, C**) and fat bodies (**D, E**) stained for pS437-Acn (green) and DNA from $w^{1118}$ and $nil^1$. (**F–J**) Projection of confocal micrographs of larval eye discs stained for pS437-Acn (green) and DNA from GMR-Gal4/+ (**F, F'**), $nil^1$ (**G, G'**), GMR-Gal4, UAS-Nil$^{WT}$; $nil^1$ (**H, H'**), GMR-Gal4, UAS-hPPM1B; $nil^1$ (**I, I'**), GMR-Gal4, UAS-Nil$^{D231N}$; $nil^1$ (**J, J'**). (**K–O**) Projection of confocal micrographs of larval fat bodies from $nil^{Ty1-G4}$ all stained for Ty1(green) and Acn (magenta) (**K- K''**), YFP-Rab5 (magenta) (**L-L''**), Rab7 (magenta) (**M-M''**), Arl8 (magenta) (**N- N''**), or Atg8a (magenta) (**O-O''**). Arrowheads in M-M'' indicate colocalization of Nil-Ty1 with Rab7 in cytosolic punctae. Notice that projections in L–O represent apical regions largely excluding nuclei. Scale bar in B and D is 40 µm for B-E and in F and K is 20 µm for F-O. Genotypes are listed in *Supplementary file 3*.

The online version of this article includes the following source data and figure supplement(s) for figure 2:

**Figure supplement 1.** Effects of Nil overexpression depend on its phosphatase activity.

**Figure supplement 2.** Nil expression in larval and adult tissues.

**Figure supplement 2—source data 1.** Raw Western blot data with molecular weight markers for *Figure 2—figure supplement 2O* from lysates of adult male flies of $nil^{Ty1-G4}$ and appropriate control probed for Ty1 and Hook.

To examine subcellular localization of the Nil phosphatase, we stained $nil^{Ty1-G4}$ animals using anti-bodies against the Ty1 tag. We could barely detect Ty1-tagged Nil phosphatase in eye discs of wandering third instar larvae (*Figure 2—figure supplement 2A, B*). However, Nil phosphatase was abundant in nuclei of third instar larval fat bodies (*Figure 2—figure supplement 2C-E*), consistent with its role in regulating phosphorylation of the primarily nuclear Acn protein (*Haberman et al., 2010*; *Nandi et al., 2014*; *Nandi et al., 2017*). Costaining of third instar larval fat bodies of $nil^{Ty1-G4}$ larvae indicated expression of endogenous Nil phosphatase in Acn-positive nuclei (*Figure 2K*). Moreover, cytosolic punctae positive for the Nil phosphatase (*Figure 2—figure supplement 2C-E*) prompted us

to examine its possible localization to endolysosomal compartments. We compared Nil localization to that of the early endosomal marker YFP-Rab5 (*Dunst et al., 2015*), but we found no colocalization of Ty1-positive Nil phosphatase punctae with YFP-Rab5 (*Figure 2L*). We further costained *nil*^[Ty1-G4] larval fat bodies with antibodies against Ty1 and Rab7, Arl8, or ATG8a. Rab7 marks late endosomes (*Numrich and Ungermann, 2014*), Arl8 lysosomes (*Rosa-Ferreira et al., 2018*), and ATG8a autophagosomes and early autolysosomes (*Klionsky et al., 2016*). We observed many of the prominent Ty1-stained Nil phosphatase punctae to colocalize with Rab7 (arrowheads in *Figure 2M*) or to be adjacent to Arl8-marked lysosomes and Atg8a-marked autophagosomes/autolysosomes (*Figure 2N and O*). This localization suggested a possible involvement of the Nil phosphatase in regulating components of endolysosomal or autophagic trafficking (2K–O, *Figure 2—figure supplement 2*) consistent with our previously described role of Acn in this pathway (*Nandi and Krämer, 2018*).

## Phosphorylation of Acn serine 437 in *Nil*[1] animal elevates basal autophagy

Increased Acn-S437 phosphorylation elevates the level of basal, starvation-independent autophagy (*Nandi et al., 2017*). We therefore tested whether *nil*[1] null animals exhibited increased levels of autophagy. Consistent with increased autophagy, *nil*[1] displayed increased ratio of lipidated Atg8a-II to Atg8a-I (*Figure 3A and B*). Furthermore, we examined endogenous Atg8a in eye and antennal discs of fed wandering third instar larvae. We observed higher numbers of Atg8a-positive punctae in *nil*[1] imaginal discs compared to wild type, possibly indicating elevated levels of autophagy (*Figure 3C–H*). Larval fat bodies are a well-established model in *Drosophila* for investigating autophagy (*Rusten et al., 2004*; *Scott et al., 2004*). In fat bodies of fed 96 hr larvae, we observed few Atg8a-positive punctae but their number increased upon a 4 hr amino acid starvation (*Figure 3I, M, Q and R*). However, numerous ATG8a-positive structures were found in fed *nil*[1] larval fat bodies (*Figure 3J and Q*) and their abundance further increased on starvation (*Figure 3N and R*).

The increased number of ATG8a punctae in *nil*[1] animals may either represent a failure of autophagosomes to fuse with lysosomes, or an enhanced autophagy induction and flux. To distinguish between these possibilities, we inhibited lysosomal acidification and degradation with chloroquine (CQ) (*Mauvezin et al., 2014*). For starved wild-type and *nil*[1] larval fat bodies, CQ treatment further elevated ATG8a staining after starvation, consistent with enhanced autophagy flux in these starved tissues (*Figure 3O, P and R*). Most importantly, treating fed 96 hr larval *nil*[1] fat bodies with CQ significantly enhanced the number of ATG8a-positive punctae demonstrating an elevated autophagic flux (*Figure 3K, L and Q*). Additionally, in heads of adult *nil*[1] animals we did not observe accumulation of p62/Ref(2)P, consistent with active autophagy flux (*Figure 3—figure supplement 1*). Enhanced autophagic flux in *nil*[1] animals in which phosphorylation of Acn S437 is robustly elevated is consistent with our previous work, that showed by several methods that phosphorylation of this residue elevates autophagic flux (*Nandi et al., 2017*).

## Cdk5-p35 kinase complex triggers Acn S437 phosphorylation in *Nil*[1] animals

PPM1-type serine-threonine phosphatases can negatively regulate stress-responsive MAPK kinase cascades by directly dephosphorylating and thereby deactivating MAP kinases or MAPK activating kinases (*Hanada et al., 1998*; *Takekawa et al., 1998*). Therefore, we examined a possible role of the Nil phosphatase in regulating the activity of the MAPK cascades and their possible involvement in phosphorylating Acn-S437 in *nil*[1] mutants. We examined phosphorylation of three MAP kinase family members: Extracellular-signal-regulated kinase (ERK), Stress-activated protein kinases / Jun amino-terminal kinases (SAPK/JNK), and p38b MAPK in larval eye discs of *nil*[1] animals. We did not observe changes in phosphorylation of ERK or SAPK/JNK in *nil*[1] eye-antennal discs compared to wild type (*Figure 4A–D*). p38b MAPK exhibited elevated phosphorylation in some undifferentiated cells anterior to the morphogenetic furrow, but not in the developing photoreceptor neurons of *nil*[1] eye disc compared to wild type (*Figure 4E–F*). Furthermore, we have demonstrated earlier that phosphorylation of Acn-S437 remains unchanged in *p38b* mutant eye discs compared to wild type (*Nandi et al., 2017*), arguing against an involvement of p38b MAPK in regulating Acn phosphorylation at serine 437 in *nil*[1] animals.

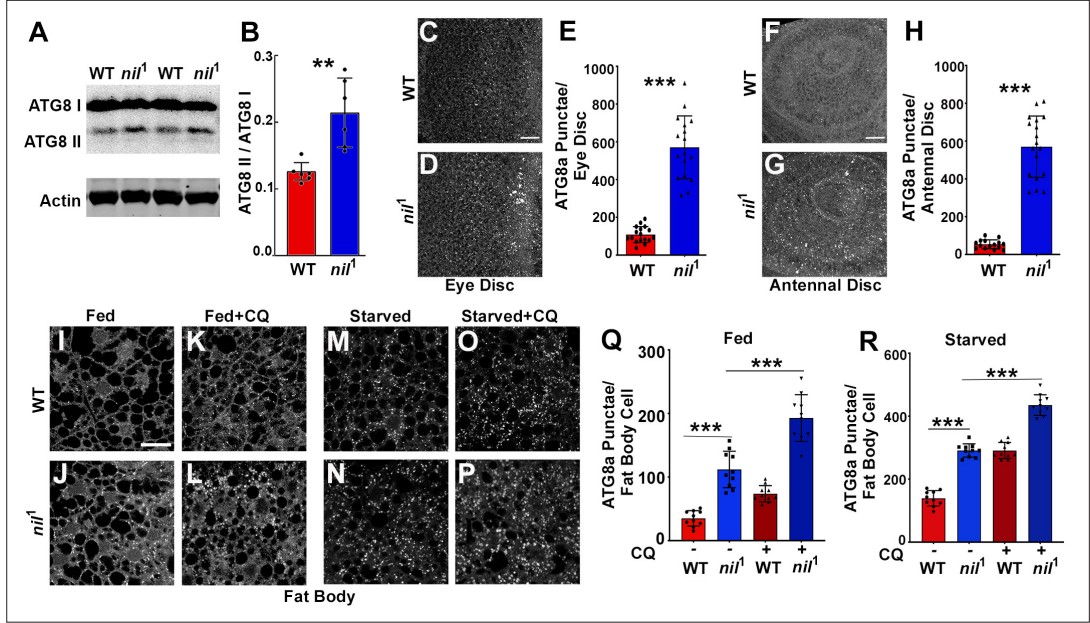

**Figure 3.** Loss of Nil enhances autophagic flux. (**A–B**) Western blot of lysates from adult heads of $w^{1118}$ and $nil^1$ probed for ATG8a (**A**). Quantification of ATG8a-II to ATG8a-I ratio from Western blots as in (**A**). Data are from six different lysates from three experimental repeats (**B**). Normality of data distributions was assessed using the Kolmogorov-Smirnov test and statistical significance using two-tailed paired t test (p=0.006). Bar graphs show mean ± SD. (**C–E**) Projection of confocal micrographs of fed $w^{1118}$ and $nil^1$ larval eye discs (**C, D**) stained for Atg8a. (**E**) Quantification of Atg8a punctae in eye discs (**C, D**) of fed $w^{1118}$ and $nil^1$. Data are from 15 larvae taken from three experimental repeats. Normality of data distributions was assessed using the Kolmogorov-Smirnov test and statistical significance using two-tailed unpaired t test (p<0.001). Bar graphs show mean ± SD. (**F–H**) Projection of confocal micrographs of fed $w^{1118}$ and $nil^1$ antennal discs (**F, G**) stained for Atg8a. (**H**) Quantification of Atg8a punctae in antennal discs (**F, G**) of fed $w^{1118}$ and $nil^1$. Data are from 15 larvae from three experimental repeats. Normality of data distributions was assessed using the Kolmogorov-Smirnov test and statistical significance using two-tailed unpaired t test (p<0.001). Bar graphs show mean ± SD. (**I–P**) Projection of confocal micrographs encompassing 6–8 cells of $w^{1118}$ and $nil^1$ larval fat bodies aged 96 hr after egg laying, either fed (**I–L**) or amino acid starved for 4 hr in 20% sucrose solution (**M–P**) stained for Atg8a. Larvae were matched for size. To assess autophagic flux, for panels K, L and O, P lysosomal degradation was inhibited with chloroquine (CQ). (**Q–R**) Quantification of Atg8a punctae in fed (**Q**) or amino acid starved (**R**) $w^{1118}$ and $nil^1$ larval fat bodies untreated or treated with CQ averaged from six to eight cells each per fat body. Data are from 10 larvae from three experimental repeats. Normality of data distributions was assessed using the Kolmogorov-Smirnov test and statistical significance using one-way analysis of variance with Tukey's correction for multiple comparisons. Bar graphs show mean ± SD. ***p<0.001. Scale bar in C, F, I is 20 µm for C–D, F–G, I–P. Genotypes are listed in *Supplementary file 3*.

The online version of this article includes the following source data and figure supplement(s) for figure 3:

**Source data 1.** Raw Western blot data with molecular weight markers for *Figure 3A* from lysates of adult heads of $w^{1118}$ and $nil^1$ probed for ATG8a and Actin.

**Figure supplement 1.** p62/ Ref(2)P is not accumulated in adult heads of $nil^1$ animal.

**Figure supplement 1—source data 1.** Raw Western blot data with molecular weight markers for *Figure 3— figure supplement 1A* from lysates of adult heads of $w^{1118}$ and $nil^1$ probed for Ref(2)P and Actin.

By contrast, the Cdk5-p35 kinase complex can directly phosphorylate Acn-S437 (*Nandi et al., 2017*). To further test whether kinases other than Cdk5-p35 contribute to elevated pS437-Acn in $nil^1$ animals, we examined Acn S437 phosphorylation in $nil^1$; $p35^{20C}$ double mutants. With the exception of dividing cells close to the morphogenetic furrow, $nil^1$; $p35^{20C}$ double mutant eye discs failed to display the pS437-Acn levels (*Figure 4J*) observed in $nil^1$ eye discs (*Figure 4I*) and instead were similar to $p35^{20C}$ mutants (*Figure 4H and J*). In cells close to the morphogenetic furrow, Acn can be phosphorylated in a Cdk5-p35 kinase-independent manner, possibly by other cyclin-dependent kinases (*Nandi et al., 2017*), as seen in the $p35^{20C}$ mutant eye discs (*Figure 4H*). The broadened stripe of pS437-Acn

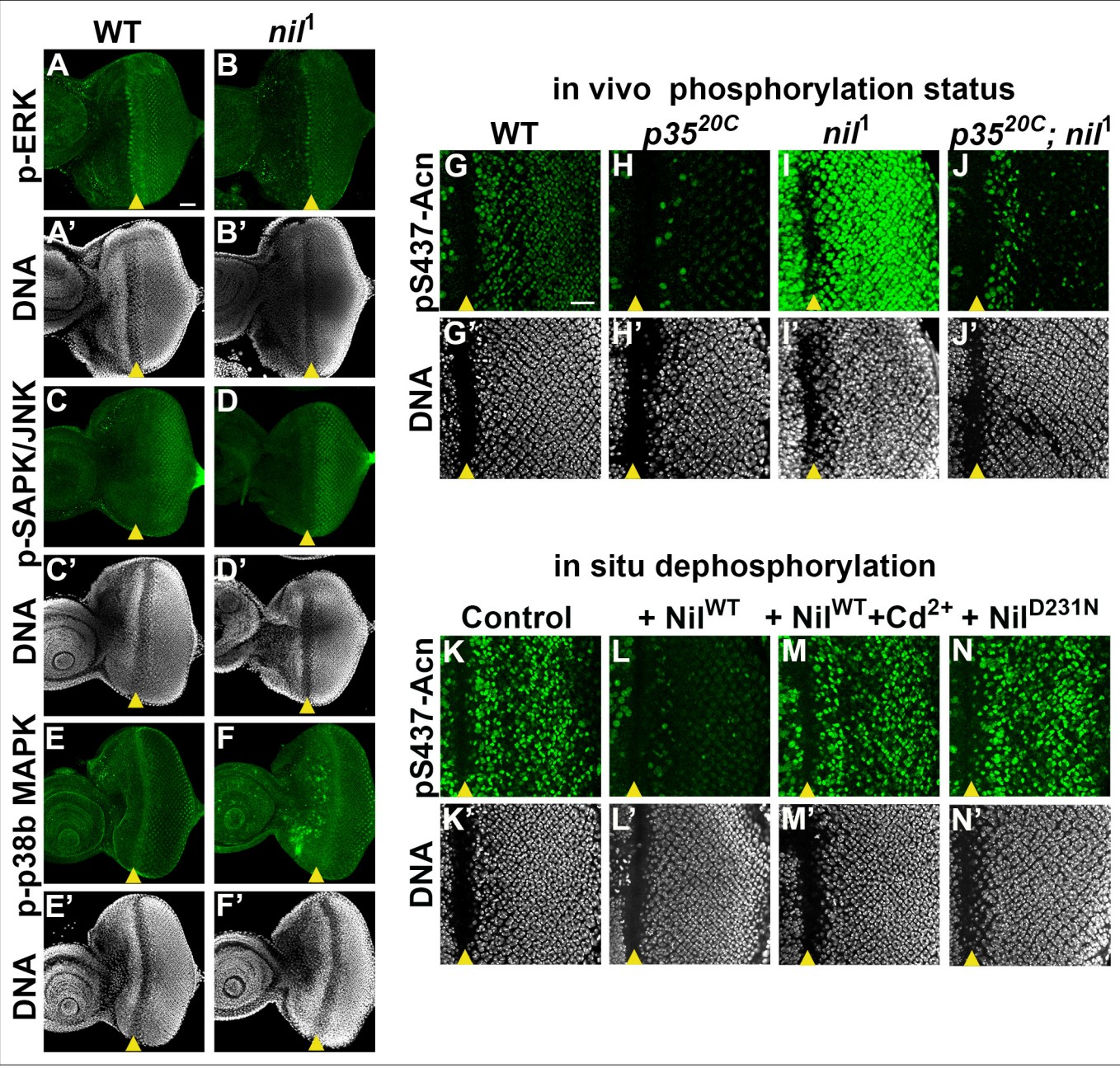

**Figure 4.** Nil dephosphorylates acinus (acn) counteracting Cdk5-p35-mediated phosphorylation. (A–F) Projection of confocal micrographs of $w^{1118}$ and $nil^1$ larval eye discs stained for p-ERK (green) and DNA (**A- B'**), p-SAPK/JNK (green) and DNA (**C-D'**), p-p38b MAPK (green) and DNA (**E-F'**). (**G–J**) Projection of confocal micrographs of larval eye discs stained for pS437-Acn (green) and DNA from $w^{1118}$ (**G, G'**), $p35^{20C}$ (**H, H'**), $nil^1$ (**I, I'**), $p35^{20C}$; $nil^1$ (**J, J'**). (**K–N**) Projection of confocal micrographs of eye discs from Acn^WT larvae stained for pS437-Acn (green) and DNA after in situ dephosphorylation without addition (**K, K'**) or with addition of wild-type Nil (**L,L'**), wild-type Nil +100 µM $CdCl_2$ (**M,M'**), or inactive Nil^D231N (**N,N'**). Scale bar in A is 40 µm for A–F. Scale bar in G is 20 µm for G–N. Genotypes are listed in **Supplementary file 3**.

positive cells closes the furrow of $nil^1$; $p35^{20C}$ double mutant eye discs suggest that the Nil phosphatase also plays a role in reversing the phosphorylation of Acn by kinases other than Cdk5-p35. Taken together, these data indicate that the Nil phosphatase counteracts Acn-phosphorylation primarily by the Cdk5-p35 kinase complex rather than inactivating a stress-responsive MAPK.

## Nil phosphatase contributes to Cd²⁺ toxicity and neurodegenerative stress

Cd²⁺ targets the M1 metal ion binding site of mammalian PPM1 phosphatases and efficiently inhibits them (**Pan et al., 2013**). To test whether the Nil phosphatase is inhibited by Cd²⁺ as well, we developed an in situ assay. Eye discs from Acn^WT larvae were fixed and detergent treated to preserve the phosphorylation status of acn which was detected by pS437-Acn staining (**Figure 4K**). In the fixed tissue, pS437-Acn was dephosphorylated by purified Nil phosphatase (**Figure 4L**), but not by Cd²⁺-inhibited Nil (**Figure 4M**) or inactive Nil^D231N (**Figure 4N**), consistent with sensitivity to Cd²⁺ inhibition being conserved in the Nil phosphatase.

Cd²⁺-induced cytotoxicity is associated with oxidative stress (**Branca et al., 2020**), which can be reduced by elevated levels of basal autophagy (**Galati et al., 2019**; **Yun et al., 2020**). To test a possible role of the Nil phosphatase in Cd²⁺-induced cellular stress responses, we examined whether pS437-Acn levels increase upon exposure to environmental Cd²⁺. We found that eye discs from wild-type larvae grown in 100 µM Cd²⁺ displayed elevated phosphorylation of Acn at S437 with a concomitant increase in the number of ATG8a-positive punctae (**Figure 5A–D** and I).

Cd²⁺ may also effect other signaling pathways with the potential to alter autophagy. We therefore wanted to test whether elevated Acn-S437 phosphorylation is necessary for Cd²⁺-induced autophagy. For this purpose, we analyzed the effect of Cd²⁺ on basal autophagy in larvae expressing either Acn^WT or Acn^S437A under control of the Acn promoter in an *Acn* null background (**Nandi et al., 2017**). We observed an increase in ATG8a punctae in fed eye discs from Acn^WT larvae grown in 100 µM Cd²⁺ similar to wild-type animals (**Figure 5E, G and J**). By contrast, Cd²⁺ exposure failed to elevate basal autophagy in the phosphoinert Acn^S437A mutants (**Figure 5F, H and J**) indicating that Acn-S437 phosphorylation is necessary for an autophagic response to Cd²⁺ exposure. Taken together, these data suggest exposure to Cd²⁺ can elevate pS437-Acn levels and enhance basal autophagy flux by deactivating Nil phosphatase.

We further explored whether conditions influencing starvation-independent and dependent autophagy can regulate nil transcription or Nil protein levels. Starvation failed to alter nil transcription in third instar *Drosophila* larvae (**Figure 5—figure supplement 1D**). Moreover, Cd²⁺ feeding capable of enhancing starvation-independent basal autophagy did not change nil transcription in third instar larvae or adult heads (**Figure 5—figure supplement 1C**,D). Additionally, Nil protein level was not significantly altered in Cd²⁺-fed Nil^Ty1-G4 adults (**Figure 5—figure supplement 1A**,B). These data confirm deactivation of Nil phosphatase and not its altered level is critical for enhancing autophagy under cellular stress.

These findings motivated us to further investigate a possible role of the Nil phosphatase in Cd²⁺-induced cytotoxicity. We exposed wild-type and *nil*¹ flies to varying concentrations of Cd²⁺ and compared their survival. Compared to wild type, median survival time for *nil*¹ mutants increased by 2 days on exposure to 125 µM Cd²⁺, 3 days at 250 µM Cd²⁺, and 5 days at 375 µM Cd²⁺ (p=0.0061, p<0.0001, and p<0.0001, respectively; log-rank Mantel-Cox test; **Figure 6A–C**, **Figure 6—figure supplement 1B**). This increase depended on Acn-S437 phosphorylation as it was not observed in *acn*^S437A; *nil*¹ double mutants (**Figure 6A–C**). The median survival time for the double mutant was even shorter compared to wild type at 125 µM Cd²⁺ (p=0.0065, log-rank Mantel-Cox test; **Figure 6A–C**). These data suggest that increased survival of *nil*¹ animals during Cd²⁺-induced stress is primarily regulated by Acn-S437 phosphorylation and the concomitant increase in basal autophagy. Interestingly, the Nil-dependent difference in susceptibility to Cd²⁺ poisoning was confined to a narrow concentrations range: at higher concentrations (500 µM) wild type and *nil*¹ mutants were not different in their survival (p=0.48, log-rank Mantel-Cox test; **Figure 6—figure supplement 1C**), and in the absence of Cd²⁺, *nil*¹ mutants had even shorter lifespans (p<0.0001 log-rank Mantel-Cox test; **Figure 6—figure supplement 1A**). These data suggest that the elevated autophagy in *nil*¹ mutants helps the animals to cope with low levels of Cd²⁺-induced oxidative stress, but is overwhelmed at higher levels.

Cdk5-p35-mediated Acn-S437 phosphorylation alleviates proteostatic stress in *Drosophila* models of neurodegenerative diseases (**Nandi et al., 2017**). Therefore, we wondered whether loss of Nil phosphatase function may reduce neurodegenerative stress. Eye-specific expression of Huntingtin-polyQ polypeptides (HTT.Q93) results in neuronal degeneration reflected by depigmentation of the adult eye (**Figure 6D and G**, **Supplementary file 2**) as previously shown (**Xu et al., 2015**). This depigmentation phenotype is suppressed by knockdown of the Nil phosphatase (**Figure 6D**,

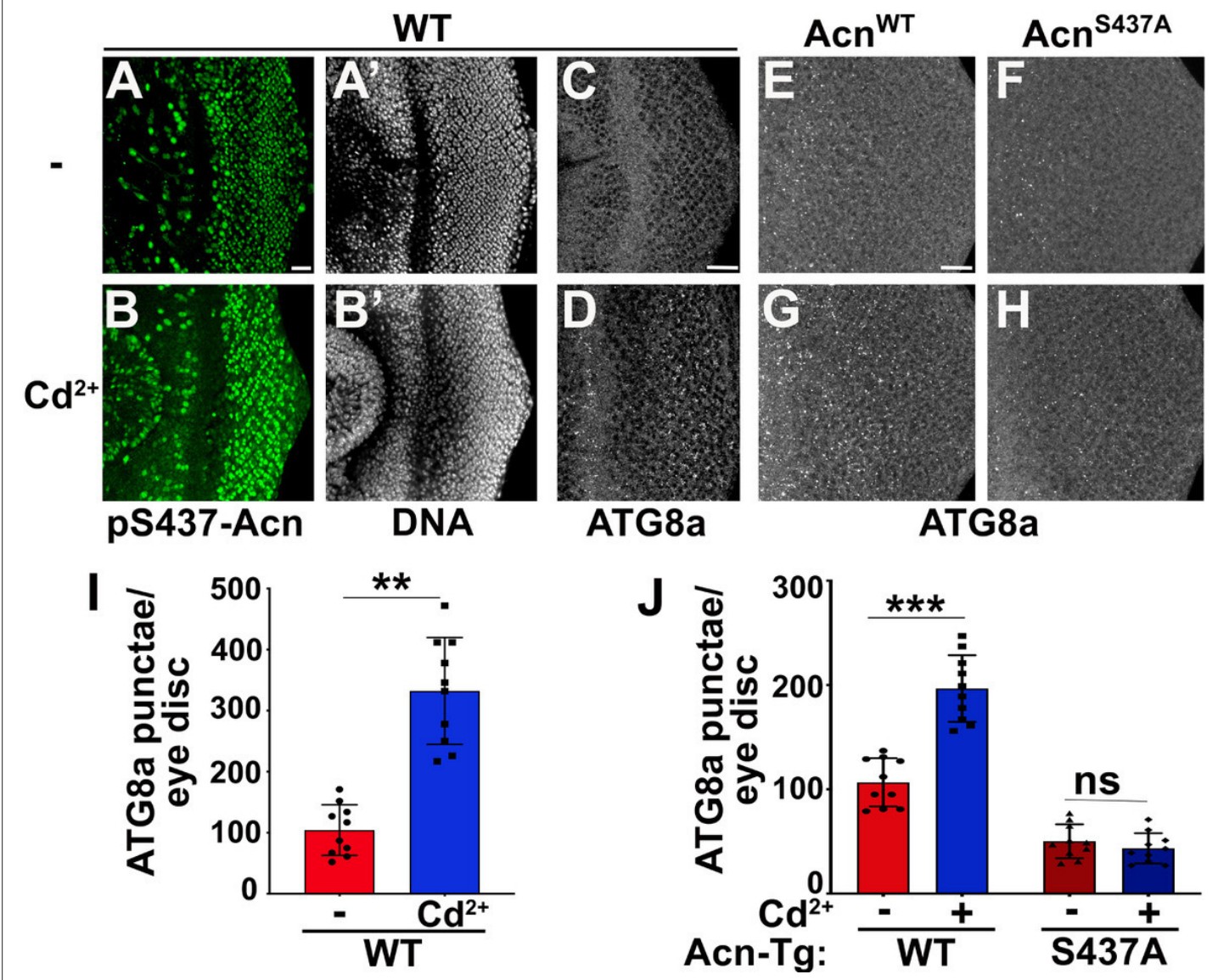

**Figure 5.** Acinus-Serine[437] phosphorylation is required for Cd[2+]-induced autophagy. (**A–B**) Projection of confocal micrographs of $w^{1118}$ larval eye discs stained for pS437-Acn (green) and DNA from either 100 μM CdCl₂ treated (**B**) or untreated (**A**) larvae. (**C–D**) Projection of confocal micrographs of fed $w^{1118}$ larval eye discs either 100 μM CdCl₂ treated (**D**) or untreated (**C**) stained for Atg8a. (**E–H**) Projection of confocal micrographs of fed Acn[WT] and Acn[S437A] larval eye discs either 100 μM CdCl₂ treated (**G, H**) or untreated (**E, F**) stained for Atg8a. (**I,J**) Quantification of Atg8a punctae in either 100 μM CdCl₂ treated or untreated $w^{1118}$ larval eye discs (**I**) or Acn[WT] and Acn[S437A] larval eye discs (**J**). Data are from 10 larvae from three experimental repeats. Normality of data distributions was assessed using the Kolmogorov-Smirnov test and statistical significance using one-way analysis of variance with Tukey's correction for multiple comparisons. Bar graphs show mean ± SD. ns, not significant; **p<0.01; ***p<0.001. Scale bar in A, C, E is 20 μm for A–H. Genotypes are listed in *Supplementary file 3*.

The online version of this article includes the following source data and figure supplement(s) for figure 5:

**Figure supplement 1.** Nil transcription or protein level does not change under conditions influencing starvation-independent or dependent autophagy.

**Figure supplement 1—source data 1.** Raw Western blot data with molecular weight markers for *Figure 5—figure supplement 1A* from adult males of $w^{1118}$, fed *nil*[Ty1-G4], and Cd[2+](250 μM) fed *nil*[Ty1-G4] probed for Ty1.

*Supplementary file 2*) but only marginally altered by overexpression of PPM1B, the human homolog of Nil (*Figure 6D*, *Supplementary file 2*), neither had an effect on pigmentation in the absence of HTT.Q93 (*Figure 6G–I*). To more directly asses the effect of Nil phosphatase on the accumulation of polyQ proteins, we stained wandering larval eye discs for polyQ proteins. GMR-driven expression of HTT.Q93 resulted in accumulation of polyQ in eye discs a few rows posterior to the furrow (*Figure 6J,*

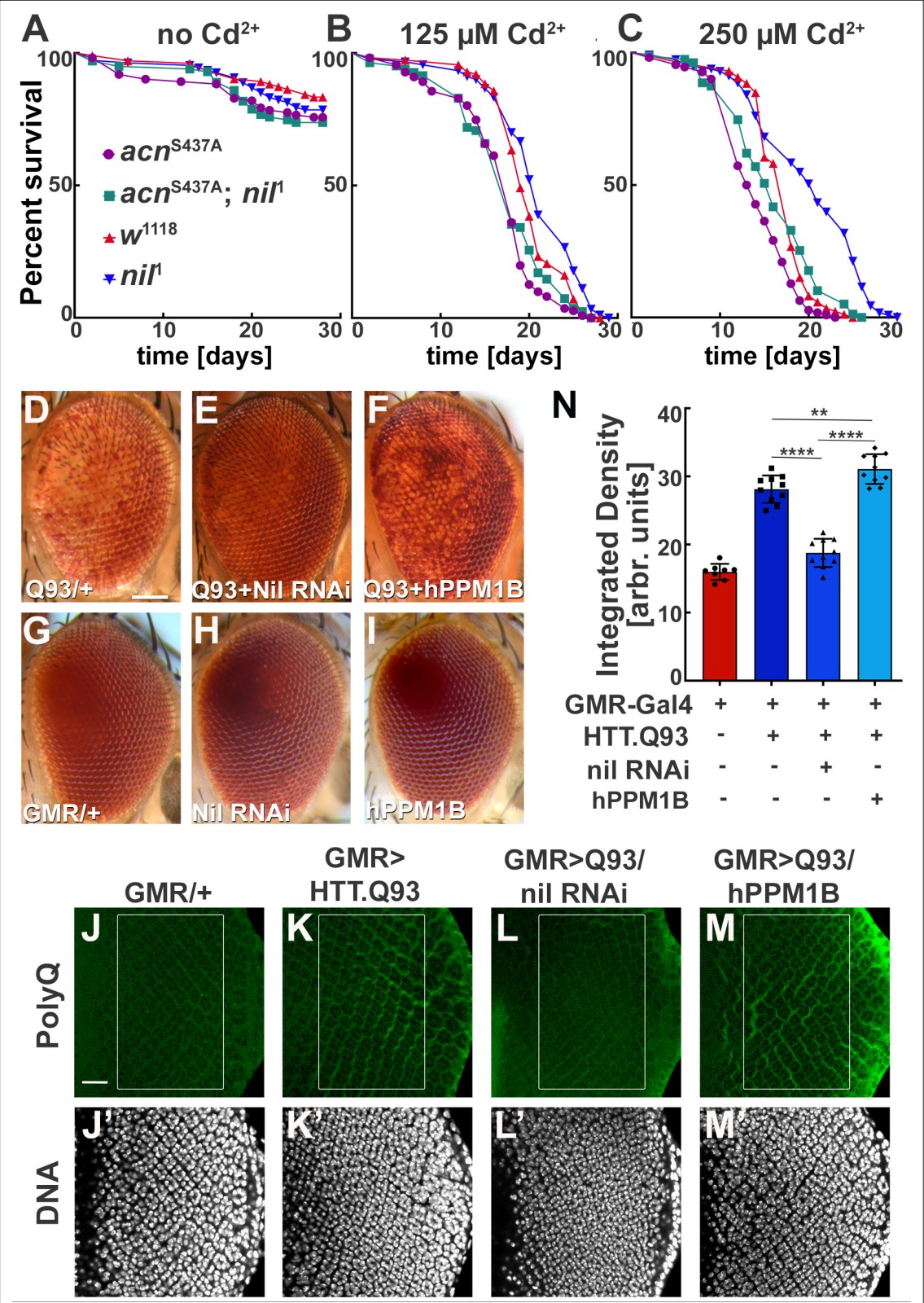

**Figure 6.** Loss of Nil function provides partial protection against Cd²⁺ poisoning and proteostasis stress. (**A–C**) Survival curves for *w*¹¹¹⁸, *nil*¹, *acn*^S437A, and *acn*^S437A; *nil*¹ adult male flies either untreated (**A**) or treated with 125 µM CdCl₂ (**B**), 250 µM CdCl₂ (**C**). N was at least 99 (**A**), 82 (**B**), and 79 (**C**). Significance was assessed using log-rank Mantel-Cox test.: ns, not significant. (**D–I**) Micrographs of adult eyes in which GMR-Gal4 drives expression of UAS-HTTex1. Q93 (**D**), UAS-HTTex1.Q93+ UAS Nil RNAi (**E**), UAS-HTTex1.Q93+ UAS-hPPM1B (**F**), UAS-Nil RNAi (**H**), UAS-hPPM1B (**I**), and (**G**) represents GMR-Gal4

*Figure 6 continued on next page*

*Figure 6 continued*

control. Scale bar in D is 100 µm for D–I. (**J–M**) Projection of confocal micrographs of larval eye discs stained for PolyQ (green) and DNA from GMR-Gal4 (**J,J'**), GMR-Gal4, UAS-HTTex1.Q93 (**K,K'**), GMR-Gal4, UAS-HTTex1.Q93+ UAS Nil RNAi (**L,L'**), GMR-Gal4, UAS-HTTex1.Q93+ UAS-hPPM1B (**M,M'**). Scale bar in J is 20 µm for J–M. (**N**) Quantification of PolyQ accumulation in eye discs of the indicated genotypes from a constant area starting two to three rows posterior to the furrow containing around 100 ommatidial clusters. The Region of interest(ROI) is represented using a box in the figure panels (**J–M**). Normality of data distribution was assessed using the Kolmogorov-Smirnov test and statistical significance using one-way analysis of variance with Tukey's correction for multiple comparisons. Bar graphs show mean ± SD of integrated densities from 10 larvae taken out of three experimental repeats. **p<0.01; ****p<0.0001. Genotypes are listed in *Supplementary file 3*.

The online version of this article includes the following figure supplement(s) for figure 6:

**Figure supplement 1.** Survival curves for adult male $w^{1118}$ and $nil^1$ flies at different $Cd^{2+}$ concentrations.

**K and N**). Knocking down Nil phosphatase expression significantly reduced polyQ accumulation posterior to the furrow (***Figure 6L and N***). By contrast, overexpression of the human PPM1B phosphatase further enhanced the polyQ load (***Figure 6M and N***). This is consistent with the data above that show elevated autophagy in $nil^1$ eye discs in combination with the known role of autophagy in the clearance of protein aggregates (***Menzies et al., 2017***). Taken together, these data suggest that PPM1-type phosphatases play an important role in regulating cellular responses to $Cd^{2+}$ toxicity and neurodegenerative stress.

## Discussion

We previously identified Acn as a signaling hub that integrates multiple stress response pathways to regulate autophagy (***Nandi and Krämer, 2018***). Starvation-independent autophagy is elevated in response to Acn being stabilized either by inhibition of its caspase-3 mediated cleavage (***Nandi et al., 2014***) or by Cdk5-p35-mediated phosphorylation (***Nandi et al., 2017***). Here, we extend this concept to Nil-regulated dephosphorylation of Acn. We show that among the serine-threonine phosphatases encoded in the *Drosophila* genome the PPM-type phosphatase Nil is specifically responsible for counteracting Acn phosphorylation by the Cdk5-p35 complex, a function conserved in the human PPM1B homolog of Nil. We used three different methods to interfere with Nil function: RNAi-induced knockdown, the CRISPR/Cas9-generated $nil^1$ null allele, or $Cd^{2+}$-mediated inhibition of Nil. In agreement with our previous analysis of the effects of the phosphomimetic Acn^S437D mutant (***Nandi et al., 2017***), all three methods yielded increased pS437-Acn levels which elevated autophagic flux, as indicated by improved clearance of polyQ load in a *Drosophila* model of Huntington's disease, or the increased number of ATG8-positive autophagosomes and autolysosomes without concomitant p62/Ref(2)p accumulation. Importantly, Acn-S437 phosphorylation was necessary for $Cd^{2+}$-induced autophagic flux and the resulting improved ability to cope with $Cd^{2+}$-induced toxicity. Thus, regulation of PPM-type phosphatase function may play an underappreciated role in the regulation of quality-control autophagy.

The family of PPM-type phosphatases is represented in the genomes of eukaryotes from yeast to humans and individual family members are highly conserved across phyla (***Kamada et al., 2020***). Despite roles of these phosphatases in diverse physiological contexts, including the regulation of metabolism, cell cycle progression, immunological responses, and other stress responses, the in vivo exploration of their functions in animal models lacks behind that of their kinase counterparts. For example, the *Drosophila melanogaster* genome encodes 15 isoforms of PPM-type phosphatases (***Kamada et al., 2020***), but to our knowledge only two of them have previously been characterized in detail using null alleles. Interestingly, these studies revealed specific regulatory roles for both phosphatases: (i) pdp (encoded by the *pyruvate dehydrogenase phosphatase* gene) dephosphory-lates the Mad signal transducer and thereby negatively regulates BMP/DPP signaling (***Chen et al., 2006***), (ii) the Alphabet phosphatase, similar to its human PPM1A/B homologs, regulates responses to developmental, oxidative, or genotoxic stresses through dephosphorylation of different MAP kinases (***Baril et al., 2009***; ***Baril and Therrien, 2006***). Alphabet, among *Drosophila* phosphatases, is the one most similar to Nil (***Kamada et al., 2020***). Therefore, we tested whether Nil also affects phosphorylation of stress-activated MAP kinases and thereby indirectly alters Acn phosphorylation. We could not find evidence supporting this possibility. In $nil^1$ mutants, pS437-Acn phosphorylation still depended on Cdk5-p35 activity and levels of phosphorylated forms of ERK, Jun, or p38 kinases appeared

unchanged. Together, these findings argue against a contribution of stress-activated kinases, other than Cdk5-p35, in Nil's effect on regulating phospho-Acn levels and autophagy.

Interestingly, other PPM-type phosphatases have also been implicated in the regulation of autophagy. In yeast, the Ptc2 and Ptc3 phosphatases redundantly regulate autophagy through the dephosphorylation of Atg1 and its binding partner Atg13 (*Memisoglu et al., 2019*). In mammalian cells, genotoxic stress activates PPM1D to dephosphorylate ULK1 and activate autophagy (*Torii et al., 2016*). In both cases, phosphatase activity counteracts the mTOR-mediated inhibition of autophagy. We do not know whether the Nil phosphatase affects other autophagy-related proteins in addition to Acn but, at least in the context of cadmium-induced autophagy, a critical step is the inhibition of Acn-S437 dephosphorylation by Nil, as cadmium can no longer induce autophagy in the phosphoinert $Acn^{S437A}$ background.

Cadmium is an environmental pollutant which, unlike many other metal ions, has no known biological role. Toxic effects of elevated cadmium levels can manifest as kidney or skeletal diseases and have been linked to multiple cancers (*Templeton and Liu, 2010*; *WHO, 2020*) and neurodegenerative disorders (*Branca et al., 2020*). The effects of cadmium on autophagy appear to be complex. Our data, in agreement with previous studies (*So et al., 2018*; *Zhang et al., 2016*), show elevated cadmium to induce autophagy with resulting cytoprotective effects, and we identify the inhibition of Nil-mediated Acn dephosphorylation as a key mechanism for this induction. In other contexts, especially cancer cells, cadmium appears to inhibit autophagy (*Liang et al., 2021*), suggesting that cadmium interacts with a distinct signaling network in those cells. Interestingly, regulation of autophagy by Acn is independent of mTOR signaling. The $Acn^{S437A}$ mutation interferes with autophagy induction by cadmium or proteostasis stress, but does not block the mTOR-dependent activation of autophagy upon amino acid starvation (*Nandi et al., 2017*).

While there is now ample support for a role of Acn in regulating autophagy (*Haberman et al., 2010*; *Nandi and Krämer, 2018*; *Nandi et al., 2017*; *Orvedahl et al., 2011*), likely upstream of Atg1/ULK kinases (*Tyra et al., 2020*), the specific mechanism is not clear yet. Nil's localization to the nucleus is consistent with effects on the well-established role of *Drosophila* and mammalian Acn proteins in alternative splicing (*Deka and Singh, 2019*; *Hayashi et al., 2014*; *Michelle et al., 2012*; *Murachelli et al., 2012*; *Nandi and Krämer, 2018*; *Rodor et al., 2016*; *Schwerk et al., 2003*; *Singh et al., 2010*). Alternatively, the subset of Nil protein localizing close to endolysosomal compartments and autophagosomes points to an alternative possibility of a more direct role of phosphorylated Acn in regulating autophagic flux. Future work will be aimed at distinguishing between these possibilities.

# Materials and methods

**Key resources table**

| Reagent type (species) or resource | Designation | Source or reference | Identifiers | Additional information |
|---|---|---|---|---|
| Gene (*Drosophila melanogaster*) | Acinus (Acn) | GenBank | FLYB:FBgn0263198 | |
| Gene (*Drosophila melanogaster*) | Cyclin-dependent kinase 5 (Cdk5) | GenBank | FLYB:FBgn0013762 | |
| Gene (*Drosophila melanogaster*) | Cdk5 activator-like protein (p35) | GenBank | FLYB:FBgn0027491 | |
| Gene (*Drosophila melanogaster*) | CG6036 | GenBank | FLYB:FBgn0039421 | |
| Genetic reagent (*Drosophila melanogaster*) | $w^{1118}$ | Bloomington *Drosophila* Stock Center | BDSC:3605; FLYB:FBst0003605; RRID:BDSC_3605 | |
| Genetic reagent (*Drosophila melanogaster*) | UAS-hPPM1B | Bloomington *Drosophila* Stock Center | BDSC:76916; FLYB: FBst0076916; RRID:BDSC_76916 | |
| Genetic reagent (*Drosophila melanogaster*) | p35-/- | PMID:17368005 | Gift from Edward Giniger, NINDS, Bethesda, Maryland, USA | |

*Continued on next page*

*Continued*

| Reagent type (species) or resource | Designation | Source or reference | Identifiers | Additional information |
|---|---|---|---|---|
| Genetic reagent (*Drosophila melanogaster*) | GMR-GAL4 | Bloomington *Drosophila* Stock Center | BDSC:1104; FLYB:FBst0001104; RRID:BDSC_1104 | |
| Genetic reagent (*Drosophila melanogaster*) | vas-Cas9(X) | BestGene | BDSC:1104; FLYB:FBst0055821; RRID:BDSC_55821 | |
| Genetic reagent (*Drosophila melanogaster*) | Da-GAL4 | Bloomington *Drosophila* Stock Center | BDSC: 55851; FLYB:FBst0055851; RRID:BDSC_55851 | |
| Genetic reagent (*Drosophila melanogaster*) | UAS-HTT-exon1-Q93 (human) | PMID:11607033 | Gift from Robin Hiesinger, Free University Berlin, Berlin, Germany | |
| Genetic reagent (*Drosophila melanogaster*) | For phosphatase RNAi lines screened the information is contained in *Supplementary file 1* | | | |
| Genetic reagent (*Drosophila melanogaster*) | YFP-MYC-Rab5 | PMID:25942626 | Gift from Marcos Gonzalez-Gaitan Marcos Gonzalez-Gaitan, MPI Dresden, Germany | |
| Genetic reagent (*Drosophila melanogaster*) | UAS-mCD8-GFP | | FLYB:FBti0012686 | |
| Genetic reagent (*Drosophila melanogaster*) | 20xUAS-6xmCherry-HA | | BDSC:52267; FLYB:FBst0052267; RRID:BDSC_52267 | |
| Genetic reagent (*Drosophila melanogaster*) | *Nil*[1] | This paper | | CRISPR/ Cas9 derived, details in "Methods" and *Supplementary file 4* |
| Genetic reagent (*Drosophila melanogaster*) | *Acn*[S437A] | This paper | | CRISPR/ Cas9 derived, details in "Methods" and *Supplementary file 4* |
| Genetic reagent (*Drosophila melanogaster*) | *Nil*[Ty1-T2A-Gal4] | This paper | | CRISPR/ Cas9 derived, details in "Methods" and *Supplementary file 4* |
| Cell line (*Drosophila melanogaster*) | S2 | | FBtc0000006 DGRC Cat# 6, RRID:CVCL_TZ72 | |
| Transfected construct (*Drosophila melanogaster*) | pMT-Nil [WT]–3xFLAG-TST | This paper | | |
| Transfected construct (*Drosophila melanogaster*) | pMT-Nil [D231N]–3xFLAG-TST | This paper | | |
| Antibody | Anti-pS437-Acn (rabbit polyclonal) | Genemed Synthesis, PMID:29227247 | | 1:1,000 (IF); Ab made against the Acn peptide H I V R D P-S(p)-P A R N R A S |
| Antibody | Guinea pig anti-Acn (aa 423–599) (guinea pig polyclonal) | PMID:20504956 | | 1:1,000 (IF) |
| Antibody | Anti-Ty1 clone BB2 (mouse monoclonal) | Invitrogen | Invitrogen: MA5-23513, RRID:AB_2610644 | 1:2000 (WB), 1:500 (IF) |
| Antibody | Anti-hook (rabbit polyclonal) | PMID:8682859 | | 1:5,000 (WB) |

*Continued*

| Reagent type (species) or resource | Designation | Source or reference | Identifiers | Additional information |
|---|---|---|---|---|
| Antibody | Anti-Actin JLA20 (mouse monoclonal) | Developmental Studies Hybridoma Bank | DSHB:JLA20; RRID:AB_528068 | 1:2000 (WB) |
| Antibody | Anti-p62 (rabbit polyclonal) | PMID:22952930 | Gift from G. Juhàsz, Eötvös Loránd University, Budapest, Hungary | 1:5,000 (WB) |
| Antibody | Anti-GABARAP [EPR4805] (rabbit monoclonal) | Abcam | Abcam:ab109364; RRID:AB_10861928 | 1:200 (IF), 1:1,000 (WB) |
| Antibody | Anti-Polyglutamine-Expansion Diseases Marker Antibody, clone 5TF1-1C2 (mouse monoclonal) | EMD Millipore | EMD Millipore:MAB1574; RRID:AB_94263 | 1:1,000(IF) |
| Antibody | Anti-V5 clone E10/V4RR (mouse monoclonal) | Invitrogen | Invitrogen:MA5-1525, RRID:AB_10977225 | 1:500 (IF) |
| Antibody | Anti-GFP clone B2 (mouse monoclonal) | Santa Cruz Biotechnology | SCB:sc-9996, RRID:AB_627695 | 1:100 (IF) |
| Antibody | Antiphospho-p44/42 MAPK (Erk1/2) (D13.14.4E) (Thr202/Tyr20) (rabbit monoclonal) | Cell Signaling Technology | CST:4,370 S, RRID:AB_2315112 | 1:200 (IF) |
| Antibody | Antiphospho-SAPK/JNK (G9) (Thr183/Tyr18) (mouse monoclonal) | Cell Signaling Technology | CST:9,255 S, RRID:AB_2307321 | 1:200 (IF) |
| Antibody | Antiphospho-p38 MAPK (D3F9) (Thr180/Tyr18) (rabbit monoclonal) | Cell Signaling Technology | CST:4,511 S, RRID:AB_2139682 | 1:500 (IF) |
| Antibody | Anti-Rab7 (rabbit polyclonal) | PMID:18272590 | Gift from Akira Nakamura, RIKEN Center for Devel. Biol., Kobe, Japan | 1:3,000 (IF); |
| Antibody | Anti-Arl8 (rabbit polyclonal) | PMID:30590083 | Gift from…… | 1:300 (IF) |
| Antibody | Alexa 488- or 568- or 647 secondaries | Molecular Probes | | 1:500 (IF) |
| Antibody | IRDye 800CW and 680RD Secondary Antibodies | LI-COR Biosciences | | 1:15,000 (WB) |
| Sequence-based reagent | DNA oligonucleotides used are listed in **Supplementary file 4** | Integrated DNA Technologies | | |
| Commercial assay or kit | Revert 700 Total Protein Stain Kits | LI-COR Biosciences | LI-COR Biosciences:926–11010 | |
| Commercial assay or kit | High-Capacity cDNA Reverse Transcription kit | Applied Biosystems (now: Thermo Fischer Scientific) | Thermo Fischer Scientific:4368813 | |
| Chemical compound, drug | Tissue-Tek O.C.T. Compound | Sakura | Sakura:4,583 | |
| Chemical compound, drug | Cadmium Chloride Anhydrous | Fisher Scientific | Fischer Scientific: C10-500 | |
| Chemical compound, drug | H-XStable Prestained Protein Ladder | UBP-Bio | UBP-Bio:L2021 | |
| Chemical compound, drug | Fast SYBR Green Master Mix | Applied Biosystems | AB:4385610 | |
| Chemical compound, drug | VECTASHIELD Antifade Mounting Medium with DAPI | Vector Laboratories | Vector Laboratories:H-1200 | |

*Continued on next page*

*Continued*

| Reagent type (species) or resource | Designation | Source or reference | Identifiers | Additional information |
|---|---|---|---|---|
| Chemical compound, drug | cOmplete ULTRA Tablets, Mini, EASYpack Protease Inhibitor Cocktail | Roche | Roche:5892970001 | |
| Chemical compound, drug | Chloroquine | Sigma-Aldrich | Sigma-Aldrich:C6628-25G | |
| Software, algorithm | Imaris software | Bitplane | RRID:SCR_007370 | |
| Software, algorithm | Adobe Photoshop | Adobe | RRID:SCR_014199 | |
| Software, algorithm | ImageJ | NIH | RRID:SCR_003070 | |
| Software, algorithm | Image Studio ver 5.2 | LI-COR | RRID:SCR_015795 | |
| Software, algorithm | Prism | Graphpad | RRID:SCR_002798 | |
| Software, algorithm | SteREO Discovery.V12 | Carl Zeiss | | |
| Software, algorithm | CZFocus | Carl Zeiss | | |
| Software, algorithm | Helicon Focus | Helicon Soft | | |

## Contact for reagent and resource sharing

Further requests for information or resources and reagents should be directed to and will be fulfilled by the Lead Contact, Helmut Kramer (helmut.kramer@utsouthwestern.edu).

## Experimental model

Fly stocks were maintained at room temperature under standard conditions. Bloomington *Drosophila* Stock Center provided Da-Gal4 (RRID:BDSC_55851), GMR-Gal4 (RRID:BDSC_1104) driver lines, $w^{1118}$ (RRID:BDSC_3605), serine-threonine phosphatase RNAi lines, UAS-hPPM1B (BS76916, RRID:BDSC_76916), UAS-mCD8-GFP (FBti0012686). Other fly strains used were $p35^{20C}$, which deletes ~90% of the *p35* coding region including all sequences required for binding to and activating Cdk5 (**Connell-Crowley et al., 2007**), a kind gift from Edward Giniger, National Institute of Neurological Disorders and Stroke, Bethesda, Maryland, YFP^MYC^-Rab5 (**Dunst et al., 2015**) and UAS-Htt-exon1-Q93 (**Steffan et al., 2001**), abbreviated UAS-Htt.Q93, was a gift from Robin Hiesinger, Free University Berlin, Berlin, Germany.

The endogenously tagged $nil^{Ty1-G4}$ gene, $nil^1$ null, and $acn^{S437A}$ mutants were generated essentially as described (**Stenesen et al., 2015**) using CRISPR/Cas9 tools available from the O'Connor-Giles, Wildonger, and Harrison laboratories (**Gratz et al., 2013**). Specifically, gRNAs (see **Supplementary file 4**) were introduced into the pU6-BbsI vector and coinjected with the appropriate template plasmid for homologous repair. Embryo injections were done by Bestgene (Chino Hills, CA), and the resulting, potentially chimeric adult flies were crossed with $w^{1118}$ flies. The vas-Cas9(X) line (BS55821, RRID:BDSC_55821) was used for CRISPR/Cas9 injections.

For the $nil^1$ null allele, the template plasmid was assembled in the pHD-DsRed backbone using approximately 1 kb PCR-amplified 5' and 3' homology arms. Potential founders were crossed to balancer stocks and resulting flies with eye-specific DsRed expression (**Bier et al., 2018**) were selected, confirmed by PCR, balanced and homozygotes collected for further analysis.

For the $Acn^{S437A}$ null allele, the template plasmid was assembled in the pHD-DsRed backbone using approximately 1 kb 5' and 3' homology arms synthesized by IDT (Integrated DNA Technologies, Inc Coralville, IA 52241, USA). Potential founders were crossed to balancer stocks and resulting flies with eye-specific DsRed expression (**Bier et al., 2018**) were selected. The DsRed cassette was removed using Cre expression, resulting lines confirmed using PCR, balanced and homozygotes collected for further analysis.

For the $nil^{Ty1-Gal4}$ allele, the template plasmid was assembled in pBS-3xTy1-T2A- Gal4. Flanking homology 5' and 3' arms of approximately 1 kb were synthesized as gBlocks by IDT with mutations in the gRNA target sites. Potential founders were crossed to flies (BS52267, RRID:BDSC_52267) containing a 20xUAS-6xmCherry-HA cassette (**Shearin et al., 2014**). Males with elevated abdominal

mCherry expression were identified and balanced. Both alleles, *nil*[1] and *nil*[Ty1–Gal4], were confirmed by sequencing of PCR products generated with one primer outside the homology arms.

Transgenic flies were generated by BestGene, Inc DNA constructs related to genomic *Acn* were generated by standard mutagenesis of a 4 kb Acn DNA fragment sufficient for genomic rescue (*Haberman et al., 2010*), confirmed by sequencing, cloned into an Attb vector, and inserted into the 96F3 AttP landing site (*Venken et al., 2006*). The UAS-Acn[WT] transgene was previously described (*Nandi et al., 2017*).

To maximize knockdown efficiency experiments with UAS-RNAi transgenes were performed at 28°C.

Life spans and survival of $Cd^{2+}$-exposed flies were analyzed as described previously (*Nandi et al., 2014*). For $Cd^{2+}$ toxicity, vials were prepared with desired concentrations of $CdCl_2$ in standard fly food. Briefly, males that emerged within a 2-day period were pooled and aged further for an additional 3 days, and then placed in demographic cages and their survival at 25°C was recorded every day. Around 100 flies were kept in each demographic cage with three replicates for each genotype. Food vials were changed every other day, and dead flies were counted and removed.

For $Cd^{2+}$ exposure of *Drosophila* larvae, 72 hr old larvae were transferred to fresh food containing the desired concentration of $CdCl_2$ at 25°C. For adults, 4–5 days old flies were exposed to $Cd^{2+}$-containing standard fly food for 4 days at 25°C.

## Histology

Eye micrographs were obtained at 72× magnification on a SteREO-microscope (SteREO Discovery. V12; Carl Zeiss) with a camera (AxioCam MRc 5; Carl Zeiss) using AxioVision image acquisition software (Carl Zeiss). Images of fly eyes are a composite of pictures taken at multiple z positions and compressed using CZFocus (Carl Zeiss) or Helicon Focus (Helicon Soft) software.

## Biochemistry

Quantitative RT-PCR was used to measure knockdown efficiencies of phosphatase RNAi lines and nil transcript levels as previously described (*Akbar et al., 2011*). In short, RNA was isolated using TRIZOL (Ambion) according to the manufacturer's protocol. High-capacity cDNA reverse transcription kit (applied biosystems) was then used to reverse transcribe 2 µg RNA with random hexamer primers. Quantitative PCR was performed using the Fast SYBR Green Master Mix in a real-time PCR system (Fast 7500; Applied Biosystems). Each data point was repeated three times and normalized for the data for ribosomal protein 49 or Act5C. Primers are listed in *Supplementary file 4*.

For immunoblot experiments, 25 adult fly heads, 5 larvae, or 5 adult flies were homogenized in 250 µl lysis buffer (10% SDS, 6 M urea, and 50 mM Tris-HCl, pH 6.8) at 95°C, boiled for 2 min, and spun for 10 min at 20,000 x*g*. An amount of 10 µl lysate were separated by SDS-PAGE, transferred to nitrocellulose membranes, blocked in 3% nonfat dry milk and probed with rabbit anti-GABARAP (EPR4805) (1:1000, Abcam, ab109364, RRID:AB_10861928) which detects ATG8a (*Kim et al., 2015*), rabbit anti-p62 (1:5000, a gift from G. Juhàsz (Eötvös Loránd University, Budapest, Hungary [*Pircs et al., 2012*]), rabbit antihook (1:5000, [*Krämer and Phistry, 1996*]), mouse antiactin (JLA20) (1:2000, Hybridoma Bank, RRID:AB_528068), and mouse anti-Ty1 clone BB2 (1:2000, Invitrogen, RRID:AB_2610644). Bound antibodies were detected and quantified using IR-dye labeled secondary antibodies (1:15,000) and the Odyssey scanner (LI-COR Biosciences) using Image Studio ver 5.2 software (LI-COR Biosciences, RRID:SCR_015795). For normalization, total protein was measured using Revert 700 Total Protein Stain Kit (LI-COR Biosciences) following manufacturer's protocol. Briefly, the nitrocellulose membrane after transfer was stained with Revert 700 Total Protein Stain, washed with Revert 700 Wash Solution, and immediately imaged in the 700 nm channel using Odyssey imaging system. Prestained molecular weight markers (HX Stable) were obtained from UBP-Bio.

## Immunofluorescence

Whole-mount larval tissues for immunofluorescence staining were set up as described previously (*Nandi et al., 2017*). Briefly, dissected tissues after fixation in periodate-lysine-paraformaldehyde were washed in 1× PBS, permeabilized with 0.3% saponin in 1× PBS (PBSS), blocked with 5% goat serum in PBSS, and stained with the specified primary antibodies: rabbit anti-pS437Acn (1:1000, *Nandi et al., 2017*), guinea pig anti-Acn (1:1000, *Nandi et al., 2014*), mouse anti-V5 clone E10/V4RR (1:500,

Invitrogen, RRID:AB_10977225), mouse anti-Ty1 clone BB2 (1:500, Invitrogen, RRID:AB_2610644), mouse anti-GFP clone B-2 (1:100, Santa Cruz Biotechnology, RRID:AB_627695), rabbit anti-Arl8 at 1:300, (*Boda et al., 2019*), rabbit anti-Rab7 (1:3000, *Tanaka and Nakamura, 2008*, a kind gift from Akira Nakamura, RIKEN Center for Developmental Biology, Kobe, Japan), rabbit antiphospho-p44/42 MAPK (Erk1/2) (Thr202/Tyr204) (D13.14.4E) (1:200, CST, RRID:AB_2315112), mouse antiphospho-SAPK/JNK (Thr183/Tyr185) (G9) (1:200, CST, RRID:AB_2307321), rabbit antiphospho-p38 MAPK (Thr180/Tyr182) (D3F9) (1:500, CST, RRID:AB_2139682), mouse anti-Polyglutamine-Expansion Diseases Marker Antibody, clone 5TF1-1C2 (1:1,000; MAB1574; EMD Millipore, RRID:AB_94263), rabbit anti-GABARAP (EPR4805) (1:200; Abcam, ab109364, RRID:AB_10861928), which detects endogenous Atg8a (*Kim et al., 2015*).

For immunostaining adult testes, male flies were anesthetized and testes were dissected and fixed in 4% paraformaldehyde and permeabilized in PBSS, blocked with 10% normal goat serum (NGS) in PBSS, and stained overnight with mouse anti-GFP clone B-2 (1:100, Santa Cruz Biotechnology, RRID:AB_627695). For head sections, female flies were anesthetized and the heads were removed and dissected in ice-cold 4% paraformaldehyde to remove the proboscis for rapid fixation. Heads were further fixed in 4% paraformaldehyde at 4°C for 1 hr followed by an overnight incubation in 25% sucrose in PBS. OCT compound (Sakura) was used to embed the heads in blocks for sectioning on a Leica CM1950 cryostat. About 20 μm sections were collected, permeabilized, blocked with 10% NGS, and incubated overnight with mouse anti-GFP clone B-2 (1:100, Santa Cruz Biotechnology, RRID:AB_627695). Next, the tissues stained with primary antibodies overnight were washed and stained with secondary antibodies conjugated to Alexa Fluor 488, 568, or 647 (1:500; Molecular Probes) and mounted in Vectashield containing DAPI (Vector Laboratories). Fluorescence images were captured with 63×, NA 1.4 or 40×, NA 1.3 or 20×, NA 0.8 plan apochromat lenses on an inverted confocal microscope (LSM 710; Carl Zeiss). Confocal Z-stacks were acquired at 1 μm step size.

For analyzing autophagy flux, 72 hr old larvae were transferred for 24 hr to fresh food containing 3 mg/ml CQ (Sigma) as described previously (*Lőw et al., 2013*) before continuing with dissections and Atg8a staining as described above.

Z-projections of three optical sections for fat body tissue and eight optical sections for eye discs and antennal discs, each 1 μm apart were used to quantify Atg8a punctae using Imaris software (Bitplane, RRID:SCR_007370). For fat bodies, the number of punctate quantified represents per fat body cell. Integrated densities for polyQ in identical areas posterior to the morphogenetic furrow of eye discs were quantified using Image J software (RRID:SCR_003070).

Digital images for display were opened with Photoshop (Adobe, RRID:SCR_014199) and adjusted for gain, contrast, and gamma settings.

All immunofluorescence experiments were repeated at least three times with at least three samples each.

## In situ dephosphorylation assay

For partial purification of recombinant Nil phosphatases, puromycin-selectable plasmids encoding C-terminally Twinstreptag-Flag-tagged Nil$^{WT}$ and Nil$^{D231N}$ proteins under control of the metallothionine promoter were transfected in S2 cells (RRID:CVCL_TZ72). Puromycin-selected pools of $6 \times 10^6$ cells were induced with 0.7 mM CuSO$_4$ for 16 hr. Cells were lysed in RIPA buffer, Nil proteins bound to MagStrep beads and washed following the manufacturer's instructions (IBA GmbH, Göttingen, Germany). Nil proteins were eluted using 50 mM Biotin in elution buffer (50 mM Tris pH 8.5, 40 mM NaCl, 0.1 mM EGTA, 1 mM DTT).

The in situ dephosphorylation assay was slightly modified from *Nandi et al., 2017*. Briefly, dissected third instar larval carcasses were fixed in periodate-lysine-paraformaldehyde, permeabilized using PBSS, and treated in 100 μl phosphatase assay buffer [40 mM MgCl$_2$, 40 mM MnCl$_2$, 50 mM Tris pH 8.5, 40 mM NaCl, 0.1 mM Ethylene glycol-bis(β-aminoethyl ether)-N,N,N′,N′-tetraacetic acid (EGTA), 1× Ethylenediaminetetraacetic acid (EDTA)-free protease inhibitor [Roche] and 1 mM Phenylmethyl-sulfonyl fluoride (PMSF)] with 50 ng of wild-type Nil phosphatase without or with 100 μM CdCl$_2$ or with 50 ng of the inactive phosphatase Nil$^{D231N}$ for 3 hr at 37°C. Following the phosphatase reaction, eye discs were washed three times in PBSS and stained for pS437-Acn as described above, mounted, and imaged as described above.

## Statistical methods

Statistical significance was determined in Prism software (Graphpad, RRID:SCR_002798) using one-way analysis of variance (ANOVA) for multiple comparisons, followed by Tukey's test and log-rank for survival assays. Normality of data distribution was assessed using the Kolmogorov-Smirnov test in Prism. We used two-way ANOVA for multiple comparisons, followed by Bonferroni's test for individual comparisons to separate effects of treatment and genetic background. Bar graphs generated from this analysis demonstrate means ± SD. For quantifications of fluorescence images at least three independent experiments were used. p Values smaller than 0.05 are considered significant, and values are indicated with one ($<0.05$), two ($<0.01$), three ($<0.001$), or four ($<0.0001$) asterisks.

## Acknowledgements

We thank members of the Krämer lab for helpful comments to the manuscript and technical assistance. We thank Drs. Edward Giniger, National Institute of Neurological Disorders and Stroke, Bethesda, Maryland, Robin Hiesinger, Free University Berlin, Berlin, Germany, the Vienna *Drosophila* Resource Center, and the Bloomington *Drosophila* Stock Center (NIH P40OD018537) for flies, and Akira Nakamura, RIKEN Center for Developmental Biology, Kobe, Japan, the Developmental Studies Hybridoma Bank at The University of Iowa for antibodies.

## Additional information

### Funding

| Funder | Grant reference number | Author |
| --- | --- | --- |
| National Eye Institute | R01EY010199 | Helmut Krämer |
| National Eye Institute | R21EY030785 | Helmut Krämer |

The funders had no role in study design, data collection and interpretation, or the decision to submit the work for publication.

### Author contributions

Nilay Nandi, Conceptualization, Investigation, Writing – original draft, Writing – review and editing; Zuhair Zaidi, Charles Tracy, Investigation, Writing – review and editing; Helmut Krämer, Conceptualization, Funding acquisition, Investigation, Supervision, Writing – review and editing

### Author ORCIDs

Nilay Nandi http://orcid.org/0000-0002-7088-4943
Charles Tracy http://orcid.org/0000-0001-7769-4950
Helmut Krämer https://orcid.org/0000-0002-1167-2676

### Decision letter and Author response

Decision letter https://doi.org/10.7554/eLife.72169.sa1
Author response https://doi.org/10.7554/eLife.72169.sa2

## Additional files

### Supplementary files

Supplementary file 1. Effect of knockdown of different phosphatases on eye roughness in a Acn gain-of-function model. All flies were raised at 28°C. Scores to calculate average roughness: normal = 1; mild = 2; rough = 3; strongly rough = 4. Positive or negative numbers indicate suppression and enhancement, respectively. Green or red colors highlight UAS-transgenes with more than 45% suppression or enhancement. Numbers in parenthesis indicated stock numbers of the Bloomington *Drosophila* stock center.

Supplementary file 2. Effect of loss and gain of phosphatase activity on eye pigmentation in a *Drosophila* Huntington's model. All flies were raised at 28°C. Scores to calculate depigmentation and roughness: Score 1: No depigmentation and no roughness. Score 2: Mild depigmentation and

no roughness. Score 3: Moderate depigmentation and roughness. Score 4: Extreme depigmentation and roughness.

Supplementary file 3. Genotypes of flies used for each figure. The table list relevant genotype for flies used in each figure.

Supplementary file 4. DNA oligonucleotides used. The table contains sequences of DNA oligonucleotides used for this study.

Transparent reporting form

### Data availability

All data generated or analysed during this study are included in the manuscript and supporting files.

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
