## [Editor Report]

The paper is of broad interest to readers focusing on quality control functions of autophagy contributing towards neuronal health. This work provides substantial new insights into the molecular mechanisms underlying the dynamic nature of the quality control function of autophagy. Building on their previous work, in the current advance the authors identify phosphatase as crucial for controlling a phospho-switch which counteracts a kinase complex dependent phosphorylation. The authors demonstrated this phospho-switch as a key integrator of multiple stress signals including. Overall the findings are interesting and important.

---

## [Decision Letter]

**Decision letter after peer review:**

Thank you for submitting your article "A phospho-switch at Acinus-Serine^437^ controls autophagic responses to cadmium exposure and neurodegenerative stress" for consideration by *eLife*. Your article has been reviewed by 2 peer reviewers, and the evaluation has been overseen by a Reviewing Editor and K VijayRaghavan as the Senior Editor. The following individuals involved in the review of your submission have agreed to reveal their identity: Gábor Juhász (Reviewer #1); Anuradha Bhukel (Reviewer #2).

Essential revisions:

Please see the reviews below. The approaches outlined and the data analyzed justify the major conclusions in the paper and the paper, as it stands, is suitable for publication in *eLife*. Our only recommendation would be to further discuss the interaction of nil with autophagy markers and if there exists a feedback loop between the two, and address the comments below.

*Reviewer #1 (Recommendations for the authors):*

I think that this work is very important and properly supported by the data so I recommend its publication in eLife. I only have a few comments for the authors to consider:

1. Why was this phosphatase named "Nilkantha" (Nil)? Please provide reasons (if any, but I suppose there is a reason).

2. The nil-Ty1 tag-(T2A cleavage site)-Gal4 knockin also allows the analysis of nil expression patterns in vivo via Gal4-mediated amplification, which I assume is more sensitive than anti-Ty1 immunostainings. Please consider including such expression data.

3. Are nil transcription (and protein) levels regulated by autophagy-influencing insults (such as starvation, proteostasis stress or cadmium exposure)? This could be interesting to know.

*Reviewer #2 (Recommendations for the authors):*

1. It is not clear from the graphs if the data were normal (necessary for use of bar graphs with mean +/- SEM. It would be beneficial to include the statistics information (what tests were use etc.)) in the figure itself.

2. For figure 4B there appears to be some bright signal at, I believe, morphogenetic furrow. Is this specific to this image or is pronounced in all the images from this group. Is might be worth showing quantifications of signal here.

3. Figure 4K, there is missing "WT" label.

4. Line 373-374 I believe it should be Figure 6O,P instead of 6I,P.

5. In figures, it might be worth using arrows to point at the signal accumulations or ROI. Sometimes it is not clear what or where to look at.

6. In figure 4 G-J' p3520C,nil1 failed to show pS437-Acn levels observed in nil1 but still has more green spots compared to p3520C alone. Is it specific to this sample or is observed in all the samples?

---

## [Author Response]

Reviewer #1 (Recommendations for the authors):I think that this work is very important and properly supported by the data so I recommend its publication in eLife. I only have a few minor comments for the authors to consider:1. Why was this phosphatase named "Nilkantha" (Nil)? Please provide reasons (if any, but I suppose there is a reason).

We renamed the CG6036 phosphatase “Nilkantha” (Nil) based on the analogy to Lord Shiva’s manifestation as “Blue Throat” in Hindu mythology. In the revised manuscript, we have now included this at the end of the Introduction (lines 95 -99):

"We renamed this phosphatase Nilkantha (Nil) meaning "Blue Throat". In Hindu mythology, Lord Shiva's throat turned blue upon drinking a poison thereby saving all the living. Similarly, our data indicate that deactivation of the CG6036 phosphatase (Nil) by Cadmium poisoning is critical for coping with the cellular stress it causes."

2. The nil-Ty1 tag-(T2A cleavage site)-Gal4 knockin also allows the analysis of nil expression patterns in vivo via Gal4-mediated amplification, which I assume is more sensitive than anti-Ty1 immunostainings. Please consider including such expression data.

Following this suggestion, we expressed UAS-mCD8 GFP using the nilTy1-Gal4 driver. We observed a wide spread low-level expression in adult brain, eye discs and larval fat bodies. Elevated expression levels were observed in adult testis. These observations are consistent with expression data for CG6036 documented in Flybase.

In the revised manuscript, we have included these additional data in Figure 2 figure supplement 2 and mentioned in the Results section following lines 159.

3. Are nil transcription (and protein) levels regulated by autophagy-influencing insults (such as starvation, proteostasis stress or cadmium exposure)? This could be interesting to know.

To address this possibility, we measured nil mRNA level in Cd2+-fed (100 µM) and starved larvae and adult heads of Cd2+-fed (125 and 250 µM) flies by quantitative RT-PCR. No significant change in nil transcription was induced by starvation or exposure to Cd2+ (Figure 5 figure supplement 1 C-D).

Additionally, we analyzed Nil protein level in NilTy1-Gal4 adult flies with or without exposure to 250 µM Cd2+ using Western blotting for Ty1-tagged Nil. We did not observe any significant change in Nil-Ty1 levels (Figure 5 figure supplement 1 A-B).

Therefore, conditions directly influencing starvation-independent basal autophagy or starvation-dependent autophagy do not regulate expression or protein level of nil.

In the revised manuscript, we mention these findings in the results section following line 247.

Reviewer #2 (Recommendations for the authors):1. It is not clear from the graphs if the data were normal (necessary for use of bar graphs with mean +/- SEM. It would be beneficial to include the statistics information (what tests were use etc.)) in the figure itself.

In the revised manuscript, we specifically indicate in all relevant figure legends that Normality of data was confirmed using the Kolmogorov-Smirnov test and included other statistical information, including the number of experiments and the statistical tests used.

2. For figure 4B there appears to be some bright signal at, I believe, morphogenetic furrow. Is this specific to this image or is pronounced in all the images from this group. Is might be worth showing quantifications of signal here.

This signal of p-ERK at morphogenetic furrow in the original Figure 4B was observed in this example only and not consistent in other examples imaged. In the new Figure 4B, we include a more representative example of p-ERK staining for *nil*^1^ eye discs, in line with our observation that there was no consistent elevation of p-ERK staining in *nil*^1^ eye discs. For clarification, in the revised Figure we now also indicated the location of the morphogenetic furrow in these images.

3. Figure 4K, there is missing "WT" label.

For clarification, we now added the label “Control” to the Figure 4K in which no phosphatase had been added during the in-situ dephosphorylation reaction.

4. Line 373-374 I believe it should be Figure 6O,P instead of 6I,P.

Thanks for catching this, in the revised Figure 6 the specific panels are now 6M,N.

5. In figures, it might be worth using arrows to point at the signal accumulations or ROI. Sometimes it is not clear what or where to look at.

We included a box in Figures 6J-M to indicate the ROI of the eye disc in which we quantified PolyQ load.

6. In figure 4 G-J' p3520C,nil1 failed to show pS437-Acn levels observed in nil1 but still has more green spots compared to p3520C alone. Is it specific to this sample or is observed in all the samples?

This is observed in all samples and we address this issue now in the result section following line 219 of the revised manuscript:

"In cells close to the morphogenetic furrow, Acn can be phosphorylated in a Cdk5-p35 kinase independent manner, possibly by other cyclin-dependent kinases (Nandi et al., 2017), as seen in the *p35*^20C^ mutant eye discs (Figure 4 H). The broadened stripe of pS437-Acn positive cells close the furrow of *nil*^1^; *p35*^20C^ double mutant eye discs suggests that the Nil phosphatase also plays a role in reversing the phosphorylation of Acn by kinases other than Cdk5-p35."